# Honey Bee Pathogen Prevalence and Interactions within the Marmara Region of Turkey

**DOI:** 10.3390/vetsci9100573

**Published:** 2022-10-17

**Authors:** Christopher Mayack, Haşim Hakanoğlu

**Affiliations:** 1Molecular Biology, Genetics, and Bioengineering, Faculty of Engineering and Natural Sciences, Sabancı University, İstanbul 34956, Turkey; 2Department of Entomology, Louisiana State University Agricultural Center, Baton Rouge, LA 70803, USA

**Keywords:** *Varroa* mites, RNA-seq, bee viruses, stonebrood, bee health, European foulbrood, American foulbrood, *Nosema*, chalkbrood, pathogen interactions

## Abstract

**Simple Summary:**

The health of bees is suspected to be low in Turkey due to the lower amounts of bee products that they produce per colony. Regular monitoring of bee diseases in Turkey has been lacking. We sampled bees from 115 colonies across five different locations to determine which pathogens are present and how much of each pathogen was in a bee colony. We found that the *Varroa* mite is widespread and consistently found in 90% of the bee colonies. We also found that pathogens normally transmitted by this parasite was also present in nearly 100% of the colonies sampled. We therefore concluded that the presence of *Varroa* mites is central to the decline in bee health and that management practices targeted for this parasite will most likely improve bee health in Turkey. In addition, three bee viruses are interacting with one another that influence the susceptibility to a more deadly variant responsible for colony death. Therefore, these viral interactions should be considered in the future to devise effective ways to improve honey bee health.

**Abstract:**

Beekeeping has yet to reach its full potential in terms of productivity in Turkey where it has a relatively large role in the economy. Poor colony health is suspected to be the reason for this, but comprehensive disease monitoring programs are lacking to support this notion. We sampled a total of 115 colonies across five different apiaries throughout the Marmara region of Turkey and screened for all of the major bee pathogens using PCR and RNA-seq methods. We found that *Varroa* mites are more prevalent in comparison to *Nosema* infections. The pathogens ABPV, DWV, KV, and VDV1 are near 100% prevalent and are the most abundant across all locations, which are known to be vectored by the *Varroa* mite. We therefore suspect that controlling *Varroa* mites will be key for improving bee health in Turkey moving forward. We also documented significant interactions between DWV, KV, and VDV1, which may explain how the more virulent strain of the virus becomes abundant. ABPV had a positive interaction with VDV1, thereby possibly facilitating this more virulent viral strain, but a negative interaction with *Nosema ceranae*. Therefore, these complex pathogen interactions should be taken into consideration in the future to improve bee health.

## 1. Introduction

Honey bees provide an indispensable service to the ecosystem as pollinators. Pollination is critical for crop production, maintenance of the ecosystem, wild plant reproduction, and food safety. Honey bees are also a major source of honey, as well as beeswax, propolis, and royal jelly. The first domesticated bee species is the western honey bee, (*Apis mellifera*). It is the most commonly managed and economically valuable pollinator in the world [1,2]. *Apis mellifera* is native to Europe, Asia (including the Middle East), and Africa, and was later introduced to other continents. Turkey, a bridge between Europe and Asia, encompasses a diverse ecosystem with diverse organisms including, but not limited to, honey bees. Examples representing the diverse pool of honey bee species found in Turkey are *A. m. anatoliaca*, *A. m. caucasica*, *A. m. meda*, and *A. m. syriaca* [3,4]. The sales of honey and beeswax alone contributed approximately $584 million to the Turkish economy in 2021 [5]. Moreover, the contribution of pollination by bees is estimated to be 10–15-fold higher than what is generated from beekeeping [6]. Turkey reportedly has more than 8 million beehives [7], ranking third in the world in terms of the number of beehives after India and Mainland China [7]. In addition, Turkey ranks second globally in honey production, producing approximately 100,000 tons [7]. However, it is believed that beekeeping in Turkey has yet to reach its full potential as its honey production per colony is substantially lower than countries such as Canada, Mainland China, Brazil, and the US [7]. This might be, among other factors, due to a decline in bee health in Turkey [8].

There are many factors attributed to the worldwide decline in bee health, but the main stressors include parasites, pesticide exposure, loss of foraging habitat, and poor nutrition [9,10,11]. More recently, synergistic effects from sub-lethal exposure to a number of pesticides have been linked to higher disease prevalence in honey bee colonies [12] and not all pesticides necessarily have the same effect across bee species [13]. Moreover, not previously recognized environmental pollutants have been found to be highly associated with different diseases in bee colonies [14]. Therefore, the particular role diseases and pesticide exposure has played in the recent decline of bee health has been difficult to distinguish from one another [15], consequently regular monitoring is key to understanding the role of each stressor in the most recent decline in bee health.

In the US, the number of beehives has declined from 4.6 million in 1970 to 2.8 million in 2019, a 40% decline (FAO, 2019). In Europe, the number of beehives declined from 2.1 million in 1970 to 1.6 million in 2019, a 24% decline [7]. In contrast, Turkey has witnessed a constant increase in the number of beehives, going from 1.8 million in 1970 to 8.1 million in 2019, a 450% increase [7]. However, as mentioned previously, Turkey has yet to reach its full potential in terms of beekeeping productivity relative to other major beekeeping countries. How Turkish honey bees are being affected from a variety of stressors remains unclear as there are no extensive regular disease monitoring programs [16]. Moreover, a complete viral analysis from RNA-seq data has been lacking from most pathogen studies worldwide [17]. However, bee diseases are expected to play a large role in the decline of bee health in Turkey as they have contributed to significant honey bee losses worldwide.

Therefore, we have sampled 115 honey bee colonies across five different locations throughout the Marmara region of Turkey to comprehensively document in detail the prevalence of all major honey bee pathogens. *Varroa destructor*, all 14 known bee viruses, the microsporidian fungi *Nosema* spp. (causative agent of *Nosema* disease), *Aspergillus* spp. (causative agents of stonebrood disease), *Ascosphaera apis* (causative agent of chalkbrood disease), bacterial foulbrood diseases *Paenibacillus larvae* (causative agent of American foulbrood disease) and *Melissococcus plutonius* (causative agent of European foulbrood disease), were all screened for as these have been identified to play a major role in previous honey bee colony losses [18]. Here, we aim to comprehensively assess prevalence and pathogen loads, and their interactions, throughout the Marmara region of Turkey, to gain an understanding of their potential impact on honey bee health. 

## 2. Materials and Methods

### 2.1. Sample Collection

Samples were collected in two batches from the brood box of standard Langstroth hives, the first batch was sampled in December 2020, from 19 colonies, located on Marmara Island (Island, N = 19). The second batch came from 96 colonies that were sampled across five different sites in July 2021: Marmara Island (Island), Karacabey, Mustafakemalpasa (MKP), Cinarcik, and Yalova, throughout the Marmara region of Turkey (Table 1). Prior to sample collection, varroa mite infestation level was determined using the sugar roll method [19]. The method involves placing around 300 collected nurse bees (1 cup) in a mason jar where they are coated with powdered sugar for 2 min. The jar is then shaken vigorously for 3 min, causing the mites to dislodge and fall through a mesh screen for collection on a white paper plate. The mites are then counted to determine the *Varroa* mite load with a 94% sensitivity, that is, the method will be able to identify 94% of *Varroa* mite infested colonies [19]. Each batch of collected bees were placed in wooden cages, where each cage consisted of 150–300 bees, and these were transported to the laboratory. Upon arrival, they were flash-frozen in liquid nitrogen and then stored at −80 °C. 

### 2.2. Sample Processing

Flash-frozen honey bees were then transferred into one or more 50 mL falcon tubes (each tube can house up to approximately 150 honey bees) and were either homogenized immediately or stored at −80 °C prior to homogenization. Whole-bee homogenates were made by macerating 100–150 frozen honey bees per sampled colony in 15 mL of DEPC treated water using the COVIDien PrecisionTM Disposable Tissue Grinder System (Medtronic, Dublin, Ireland). The homogenization was done in three rounds for each 50 mL falcon tube; each round consisted of macerating the content of one third of a tube in 5 mL of DEPC-treated water. 50 mL Falcon tubes were used to collect 15 mL of liquid from macerated honey bees. A total of 6 aliquots, of 150 μL, were made from each 15 mL of honey bee homogenate. Each aliquot was used to screen for one of the 5 pathogens that infect honey bees, and the remaining aliquot was allocated for RNA extraction for viral screening using RNA-seq. Primers for each PCR pathogen screening method can be found in Appendix A.

### 2.3. Pathogen Screening

#### 2.3.1. Stonebrood (SB) Screening

Genomic DNA for SB screening was extracted from whole-bee homogenate aliquots using a custom-made lysis buffer (300 μL of buffer containing 200 mM Tris-HCl (pH 7.5), 25 mM EDTA, 0.5% *w*/*v* SDS, and 250 mM NaCl per 150 μL of homogenate) followed by a standard phenol:chloroform (1:1) extraction [20]. Assessment of DNA yield and purity was done using the NanoDrop 1000c spectrophotometer (Thermo Fisher Scientific, Waltham, MA, USA). Amplification of the β-tubulin gene of the SB-causing *Aspergillus* spp., and the ribosomal protein S5 (RpS5) gene of *Apis mellifera* was performed using PCR. RpS5 is a honey bee house-keeping gene that is expressed stably across different tissues and seasons [21,22], thus allowing for monitoring of extraction failures or PCR amplification inhibition. All PCR amplifications were performed using 2× Taq PCR MasterMix (abm, Richmond, BC, Canada) in 25 μL reactions containing 400 nM each primer targeting either *Aspergillus* β-tubulin or *A. mellifera* RpS5. PCR conditions were as follows: 95 °C for 5 min; 35 cycles of 94 °C for 45 s, 60 °C for 45 s, and 72 °C for 1 min; and 72 °C for 6 min [20]. PCR products, a positive control, and a 100 bp Opti-DNA Marker (abm, Richmond, BC, Canada) were separated by gel electrophoresis at 100 V for 30 min using 1.5% agarose gels stained with GelRed Nucleic Acid Stain (10,000× in water) (Biotium, Fremont, CA, USA). Gel visualization was performed using the Bio-Rad Gel Doc EZ Gel Documentation System (Bio-Rad, Hercules, CA, USA).

#### 2.3.2. Chalkbrood (CB) Screening

Genomic DNA for CB screening was extracted from whole-bee homogenate aliquots using the DNeasy Plant Mini Kit (Qiagen, Hilden, Germany) according to the manufacturer’s protocol [23]. Assessment of DNA yield and purity was performed using the NanoDrop 1000c spectrophotometer (Thermo Fisher Scientific, Waltham, MA, USA). Amplification of the internal transcribed spacer (ITS) region within the nuclear ribosomal repeat unit of the fungus *Ascosphaera apis* [24], and RpS5 gene of *A. mellifera* was completed using PCR. All PCR amplifications were performed using 2× Taq PCR MasterMix (abm, Richmond, BC, Canada), in 25 μL reactions, containing 400 nM each primer, targeting either *A. apis* ITS or *A. mellifera* RpS5. PCR conditions were as follows: 94 °C for 10 min; 30 cycles of 94 °C for 45 s, 62 °C for 45 s, and 72 °C for 1 min; and 72 °C for 5 min [25]. PCR product evaluation was performed as above.

#### 2.3.3. American Foulbrood (AFB) Screening

Genomic DNA for AFB screening was extracted by heating the honey bee homogenate [26]. Briefly, 800 µL of DEPC-treated water was added to a whole-bee homogenate aliquot and centrifuged at 800× *g* for 10 min. 200 µL of suspension from each aliquot were incubated at 95 °C for 15 min with the lids open, then centrifuged at 5000× *g* for 5 min. The screening was performed using a SYBR Green-based qPCR for the amplification and detection of a region in the 16S rRNA gene of *P. larvae* [27]. Reactions were carried out using BrightGreen 2× qPCR MasterMix (abm, Richmond, BC, Canada) in 20 µL reactions containing 5 µL of supernatant from heated honey bee homogenate and 250 nM of each primer. qPCR amplifications and detections were performed using a LightCycler^®^ 480 System (Roche Diagnostics, Roche, Basel, Switzerland) with a program consisting of initial denaturation at 94 °C for 4 min, followed by 45 cycles of denaturation at 95 °C for 15 s and annealing and signal acquisition at 56 °C for 10 s [27]. All screenings involved no template negative controls and positive controls.

#### 2.3.4. European Foulbrood (EFB) Screening

Genomic DNA for EFB screening was extracted using a custom-made grinding buffer (500 µL of grinding buffer containing 0.25 g guanidine thiocyanate, 26.5 µL 1 M Tris-Cl (pH 7.6), and 26.5 µL 0.2 M EDTA per 150 µL of homogenate) followed by the DNeasy Plant Mini Kit (Qiagen, Hilden, Germany) according to the manufacturer’s protocol [28]. Assessment of DNA yield and purity was performed using the NanoDrop 1000c spectrophotometer (Thermo Fisher Scientific, Waltham, MA, USA). DNA samples with A260/230 < 1.7 were subjected to PCR amplification of *A. mellifera* RpS5 in duplicates. Products were visualized after being separated by electrophoresis at 100 V for 30 min on 2% agarose gels stained with GelRed Nucleic Acid Stain (10,000× in water) (Biotium, Fremont, CA, USA). Gel visualization was performed using the Bio-Rad Gel Doc EZ Gel Documentation System (Bio-Rad, Hercules, CA, USA). 

Samples that showed the RpS5 amplicon in both of their PCR duplicates or passed the QC using the NanoDrop 1000c spectrophotometer were subjected to EFB screening. Samples were subjected again to DNA extraction, quantity and quality assessment, and *A. mellifera* RpS5 amplification using PCR if one or none of their PCR duplicates produced RpS5 amplicons. EFB screening was performed using the BactoReal European Foulbrood Kit (Ingenetix GmbH, Vienna, Austria), a probe-based qPCR assay that detects the 16S rRNA gene of *Melissococcus plutonius*. According to the manufacturer’s protocol, reactions were carried out using TaqProbe 2× qPCR MasterMix (abm, Richmond, BC, Canada) in 20 µL, containing 5 µL of template. All screenings involved no template negative controls and positive controls supplied by the kits.

#### 2.3.5. Nosema Screening and Semi-Quantification of Nosema Infection

Genomic DNA extraction for *Nosema* screening was completed using the HBRC method that relies on a custom-made buffer (300 µL of 3 mM hexadecyltrimethylammonium bromide (CTAB), 5 mM Tris-HCl, 1 mM EDTA, and 1.1 M NaCl per 150 µL of homogenate) and proteinase K, followed by a standard phenol:chloroform (1:1) extraction [29]. Assessment of DNA yield and purity was performed using the NanoDrop 1000c spectrophotometer (Thermo Fisher Scientific, Waltham, MA, USA). Co-amplification of 16S rRNA gene of *Nosema apis* and *Nosema ceranae*, and RpS5 of *A. mellifera* was performed using PCR [29]. All PCR amplifications were performed using 2× Taq PCR MasterMix (abm, Richmond, BC, Canada) in 25 µL reactions containing 400 nM of each primer. PCR products were separated by electrophoresis at 100 V for 30 min on 1% agarose gels stained with GelRed Nucleic Acid Stain (10,000× in water) (Biotium, Fremont, CA, USA). Gel visualization was performed using the Bio-Rad Gel Doc EZ Gel Documentation System (Bio-Rad, Hercules, CA, USA). The pixel intensity of amplified bands was measured using ImageJ v1.53k [30]. The ratio of *N. apis* 16S rRNA band intensity to the *A. mellifera* RpS5 band intensity was calculated to semi-quantify the relative abundance of *N. apis* for each sample. Similarly, the ratio of *N. ceranae* 16S rRNA band intensity to the *A. mellifera* RpS5 band intensity was calculated to semi-quantify the relative abundance of *N. ceranae* for each sample.

#### 2.3.6. RNA Extraction, Quality Control (QC), and Sequencing for Viral Screening

RNA extraction was performed using the EcoPURE Total RNA Kit (ECOTECH Biotechnology, Erzurum, Turkey) according to the manufacturer’s protocol. The lysis buffer from the kit was mixed with β-mercaptoethanol in a volume ratio of 100:1. An on-column DNase I treatment step was applied during RNA extraction. Briefly, a 10 µL reaction containing 1 U of DNase I and 10× Reaction Buffer I (EURx, Gdańsk, Poland) was added to each column and columns were then incubated at 37 °C for 15 min for complete digestion of DNA. Preliminary assessment of total RNA yield and purity, and integrity was done using NanoDrop 1000c spectrophotometer (Thermo Fisher Scientific, Waltham, MA, USA). RNA integrity was assessed using the ‘bleach gel’ method [31]. Briefly, RNA was separated by electrophoresis for 35 min on 1% agarose gels mixed with 0.5% household bleach (6% sodium hypochlorite) prior to melting. Representative samples with different RNA concentrations, levels of degradation and intensities of 28S and 18S rRNA bands were assessed further for integrity and amount using the Agilent 2100 Bioanalyzer with the RNA 6000 Nano Kit (Agilent Technologies, Santa Clara, CA, USA), according to the manufacturer’s protocol and quantified using the QuantiFluor RNA System (Promega, Madison, WI, USA). Heat denaturation, a standard step in integrity assessment of RNA that is used to destroy secondary structures, was not performed prior to running the Agilent 2100 Bioanalyzer as it results in the fragmentation of 28S rRNA into two similarly sized fragments that migrate closely with 18S rRNA in honey bees [32]. 

RNA concentrations ranged between 60–389 ng/µL based on results from the NanoDrop 1000c spectrophotometer. Concentrations of representative samples (*n* = 26) determined using the QuantiFluor RNA System were higher than the ones determined using the NanoDrop 1000c spectrophotometer for the same samples. RNA A260/280 and A260/230 values ranged between 2.0–2.2, and 2.0–2.3, respectively. Migration profiles of RNA samples in bleach gels showed a variety of degradation levels. Samples with highly degraded RNA were re-extracted. Pure RNA with low to medium degradation, and intact 28S and 18S rRNA bands (regardless of intensity) were sent to Novogene Corporation Inc. (Cambridge, UK) for sequencing. Libraries that passed the quality control were sequenced on an Illumina NovaSeq 6000, generating ≥ 30 million paired-end 150 bp reads (PE150) per sample for 113 samples. Library construction failed for 2 out of the 115 samples so these were removed from further analysis.

#### 2.3.7. Quality Control and Pre-Processing of RNA-seq Reads for Viral Identification and Quantification

Quality Control (QC) of raw reads was completed using FastQC v0.11.7, and QC reports were summarized using MultiQC v1.12 [33]. Adapter and N base trimming of raw reads was accomplished using Cutadapt v2.5 [34] with the following options: -a agatcggaagagcacacgtctgaactccagtca -A agatcggaagagcgtcgtgtagggaaagagtgt --trim-n --max-n 0.1 m 25. The trimmed reads were aligned to the *Apis mellifera* genome assembly (Amel_HAv3.1) containing the genomes of 14 different RNA viruses that are known to infect honey bees. These viruses are acute bee paralysis virus (ABPV), NC_002548.1; aphid lethal paralysis virus (ALPV), NC_004365.1; Bee Macula-like virus, NC_027631.1; Big Sioux River virus (BSRV), NC_035184.1; black queen cell virus (BQCV), NC_003784.1; chronic bee paralysis virus (CBPV), NC_010711.1, NC_010712.1; deformed wing virus (DWV), NC_004830.2; Israeli acute paralysis virus (IAPV), NC_009025.1; Kakugo virus (KV), AB070959.1; Kashmir bee virus (KBV), NC_004807.1; Lake Sinai virus (LSV), NC_032433.1; sacbrood virus (SBV), NC_002066.1; slow bee paralysis virus (SBPV), NC_014137.1; Varroa destructor virus-1 (VDV1), NC_006494.1. The alignment was done using HISAT2 v2.2.1 [35] with --rna-strandness set to RF. 

The produced SAM files containing the alignments were sorted based on coordination, converted to BAM format, and indexed. BAM entries of reads aligned to viral genomes were extracted and reads mapped to each viral genome were counted. Sorting, conversion, indexing, extraction, and counting reads aligning only to viral genomes were completed using Samtools v1.14 [36]. The ratio of number of reads aligned to the viral genome to the total number of reads for each virus was calculated as means to quantify viral loads in samples. 

### 2.4. Statistical Analysis and Visualization

Prevalence of pathogens was calculated as the number of pathogen-positive colonies divided by the number of total sampled colonies. A pathogen-positive colony is any colony with at least one *Varroa* mite detected, in the case of *Varroa*, a pathogen-specific PCR product detected regardless of the intensity, in the case of fungal pathogens, sharp increase in fluorescent signal, in the case of bacterial pathogens, and/or more than 50 RNA-seq reads aligned to viral genomes, in the case of viruses. Differences between proportions of pathogen-positive colonies from different sampling apiary locations were assessed using Fisher’s exact test. Differences in pathogen levels between colonies from different sampling apiary locations were assessed using the Kruskal–Wallis test followed by Dunn’s test with the Benjamini-Hochberg correction for multiple pairwise comparisons. Separate analyses were made for each site to obtain an overview of the prevalence of each pathogen within each sampling site. 

Relationships between pathogens were examined using generalized mixed linear models (GLMMs). For *Varroa* mites counts, a negative binomial GLMM was built with all other pathogens as fixed predictors. For SB and AFB, binomial GLMMs were built with all other pathogens as fixed predictors. For *Nosema*, zero-inflated Gamma GLMMs were constructed with all other pathogens as fixed predictors. For viral infections, negative binomial GLMMs were built with viral read counts as the response variable and all other pathogens and an offset for total number of RNA-seq reads to account for different sequencing depths as fixed predictors. When possible, additional models were built per pathogen that also include the ABPV-BQCV interaction, DWV-KV-VDV1 interactions, or ABPV-BQCV and DWV-KV-VDV1 interaction. Variations associated with geographical locations and batches were accounted for by including them as random effects in each model. GLMMs with and without interactions were tested for differences in residual deviance using the likelihood ratio test (LRT). Insignificant differences in residual deviance between two models indicates that they fit the data similarly, and therefore the model with the least complexity (less/no interaction effects) will be chosen per pathogen. Model residuals, over-/under-dispersion, outliers, and zero-inflation were checked using a simulation-based approach via the package DAHRMa v0.4.5 [37]. Continuous predictors were standardized prior to model fitting. All GLMMs were fitted by maximum likelihood with Laplace approximation using the ‘glmer’ function for binomial GLMMs and ‘glmer.nb’ for negative binomial GLMMs of the R package lme4 v1-1.27.1 [38]. Zero-inflated Gamma models using the function ‘glmmTMB’ of the R package glmmTMB v1.1.3 [39]. Negative binomial GLMs were fitted by maximum likelihood using the ‘glm.nb’ function of the R package MASS v7.3-54 [40]. Coefficient estimates, significance and 95% confidence intervals were retrieved from models using the ‘get_model_data’ function of the package sjPlot v2.8.9 [41]. All analyses and visualizations were performed in R 4.1.0 [42]. Boxplots were created using the packages ggpubr v0.4 [43], while barplots and Model coefficient estimate plots were created using the package ggplot2 v3.3.5 [44]. The Nature Publishing Group (NPG) color palette used in plots was retrieved from ggsci v2.9 [45].

## 3. Results

### 3.1. Prevalence of Pathogens

*Varroa* mite prevalence did not significantly vary across the 5 sampling locations (Fisher’s exact test: *p* = 0.08, Figure 1). The difference in proportions of *N. ceranae* infected colonies across the different sampling locations was significant (Fisher’s exact test: *p* < 0.001, Figure 1), while *N. apis* was not detected in any of the colonies. CB was not detected in any of the colonies, while the variation of SB across the locations was significant (Fisher’s exact test: *p* < 0.01, Figure 1) The difference in proportions of AFB infected colonies across the different sampling locations was significant (Fisher’s exact test: *p* < 0.001). Because EFB prevalence was the lowest, it was dropped from further analyses. 

*Varroa* mite counts vary significantly across the different sampling locations (Kruskal–Wallis test: H = 23.41, df = 4, *p* < 0.001; Figure 2A). Similarly, *N. ceranae* levels also vary significantly across the different sampling locations (Kruskal–Wallis test: H = 72.33, df = 4, *p* < 0.001; Figure 2B). 

Chronic bee paralysis virus (CBPV), Israeli acute paralysis virus (IAPV), and slow bee paralysis virus (SBPV) were not detected in any of the sampled colonies; while Big Sioux River virus (BSRV) and Kashmir bee virus (KBV) the number of RNA-seq reads aligning to their genomes per sample is less than 50 reads. Therefore, these viruses were not analyzed further. VDV1, BQCV, DWV, and KV were found in nearly 100% of the colonies sampled and the prevalence did not significantly differ by sampling location (Figure 3). The difference in ABPV prevalence across the different sampling locations is significant (Fisher’s exact test: *p* < 0.001, Figure 3). The difference in LSV and BeeMLV prevalence across the different sampling locations was significant (Fisher’s exact test: *p* < 0.001, Figure 3). Aphid lethal paralysis virus (ALPV) was detected only in 11.76% and 5.88% of colonies located on the Marmara Island and in Yalova; while sacbrood virus (SBV) was only detected in 11.76% of colonies located in Yalova (Figure 3).

Both VDV1 and BQCV levels were not significantly different in colonies from different locations (VDV1: H = 8.73, df = 4, *p* = 0.68; BQCV: H = 5.83, df = 4, *p* = 0.21; Figure 4A,B); while DWV, KV, and ABPV levels were significantly different (DWV: H = 27.92, df = 4, *p* < 0.001; KV: H = 27.4, df = 4, *p* < 0.001; ABPV: H = 20.83, df = 4, *p* < 0.001; Figure 4C–E). Colonies located on the Marmara Island had significantly lower DWV levels compared to ones located in Karacabey (Dunn’s test: adj-*p* < 0.001, Figure 4C) and Yalova (Dunn’s test: *p* < 0.001, Figure 4C). They also had significantly lower KV levels compared to ones located in Karacabey (Dunn’s test: adj-*p* < 0.001, Figure 4D), MKP (Dunn’s test: adj-*p* < 0.01, Figure 4D), and Yalova (Dunn’s test: adj-*p* < 0.01, Figure 4D). In contrast, the same colonies have significantly higher levels of ABPV compared to ones located in MKP (Dunn’s test: adj-*p* < 0.05, Figure 4E) and Yalova (Dunn’s test: adj-*p* < 0.05, Figure 4E). Colonies located in Karacabey have significantly higher ABPV levels compared to Cinarcik (Dunn’s test: adj-*p* < 0.05, Figure 4E), MKP (Dunn’s test: adj-*p* < 0.01, Figure 4E), and Yalova (Dunn’s test: adj-*p* < 0.01, Figure 4E). The levels of the less prevalent viruses, ALPV, BeeMLV, LSV, and SBV at least two orders of magnitude less than that of VDV1, BQCV, DWV, KV, and ABPV (Figure 4F–I). Therefore, they were dropped from further analyses. 

### 3.2. Relationships between Honey Bee Pathogens

#### 3.2.1. Non-Viral Infections

For *Varroa* mites counts, SB and AFB prevalence, and *Nosema* levels, negative binomial generalized linear models (GLMMs), binomial GLMMs, and zero-inflated GLMMs were built with all other pathogens as fixed predictors, respectively. Per disease, four GLMMs were compared for their goodness of fit: the first model included the ABPV-BQCV interaction, the second model included DWV-KV-VDV1 pairwise interactions, the third model included the ABPV-BQCV interaction as well as the DWV-KV-VDV1 pairwise interactions, and fourth model did not include any interactions. Differences in residual deviance between models with and without interactions for all pathogens were not significant (Appendix A). Therefore, the models without interactions were further examined. The *Varroa*-specific GLMM (NB *Varroa* GLMM) revealed positive association between *Varroa* mite counts and SB prevalence (NB *Varroa* GLMM: SB estimate = 0.88, *p* < 0.001; Figure 5A). This relationship was also seen in the SB-specific GLMM (SB GLMM: *Varroa* estimate = 1.5, *p* < 0.01; Figure 5B). NB *Varroa* GLMM also showed a negative association between Varroa mite counts and *N. ceranae* levels (NB *Varroa* GLMM: *Nosema* estimate = −0.39, *p* < 0.05; Figure 5A). However, this relationship was not seen in the *Nosema*-specific GLMM (ziGamma *Nosema* GLMM: *Varroa* estimate = −0.13, *p* = 0.36; Figure 5D). In fact, *N. ceranae* were not associated with any pathogen (Figure 5D). Similarly, AFB prevalence was not associated with any pathogen (Figure 5C).

#### 3.2.2. Viral Infections

For each of ABPV and BQCV, two NB GLMMs were built with all other pathogens as fixed predictors and compared for their goodness of fit: the first one included DWV-KV-VDV1 pairwise interactions, while the second one included no interactions. Differences in residual deviance between models with and without interactions for all pathogens were not significant (Appendix A). Therefore, for both ABPV and BQCV, the models without interactions were examined further. As for DWV, KV, and VDV1, four GLMMs were built per virus with all other pathogens as fixed predictors and compared for their goodness of fit: the first model included the ABPV-BQCV interaction (this model did not converge for DWV), the second model included any two of the DWV-KV-VDV1 depending on the response variable, the third model included the ABPV-BQCV interaction as well as the DWV-KV-VDV1 pairwise interactions, and fourth model did not include any interactions. Differences in residual deviance between models with and without interactions for all pathogens were not significant (Appendix A). Differences in residual deviance between the model without interactions and models with KV-VDV1, DWV-VDV1, and DWV-KV were significant for DWV (LRT: χ^2^ = 18.67, df = 3, *p* < 0.001; Appendix A), KV (LRT: χ^2^ = 13.6, df = 1, *p* < 0.001; Appendix A), and VDV1 (LRT: χ^2^ = 17.16, df = 1, *p* < 0.001; Appendix A). In addition, the difference in residual deviance between the VDV1 model with all interactions and the one without any interactions was also significant (LRT: χ^2^ = 17.16, df = 1, *p* < 0.001; Appendix A), and the model with interactions was compared to the one with the DWV-KV interaction. The difference in residual deviance between these two was not significant (LRT: χ^2^ = 3.4, df = 2, *p* = 0.18). Therefore, the models with KV-VDV1, DWV-VDV1, and DWV-KV interactions were examined further for DWV, KV, and VDV1, respectively.

The ABPV-specific GLMM (NB ABPV GLMM) revealed negative association between ABPV levels and *N. ceranae* levels (NB ABPV GLMM: *Nosema* estimate = −0.7, *p* < 0.01; Figure 6A) and positive association between ABPV levels and VDV1 levels. However, the first relationship was not seen in the ziGamma *Nosema* GLMM (Figure 5D) and the second one was also not seen in the VDV1-specific GLMM (NB VDV1 GLMM: ABPV estimate = −0.02, *p* = 0.85; Figure 6E). BQCV levels were not associated with any pathogens (Figure 6B). The DWV-specific GLMM revealed positive associations between DWV levels and *Varroa* mite counts (NB DWV GLMM: *Varroa* estimate = 0.22, *p* < 0.05; Figure 6C), as well as KV (NB DWV GLMM: KV estimate = 1.56, *p* < 0.001; Figure 6C) and VDV1 levels (NB DWV GLMM: VDV1 estimate = 0.44, *p* < 0.001; Figure 6C). The first relationship was not seen in the NB *Varroa* GLMM (NB Varroa GLMM: DWV estimate = 0.23, *p* = 0.2; Figure 5A), while the last two relationships were seen in the KV-specific GLMM (NB KV GLMM: DWV estimate = 1.19, *p* < 0.001; Figure 6D) and the VDV1-specific GLMM (NB VDV1 GLMM: DWV estimate = 0.77, *p* < 0.001; Figure 6E), respectively. The NB DWV GLMM also revealed a negative association between DWV levels and the interaction between KV and VDV1 levels (NB DWV GLMM: KV × VDV1 estimate = −0.98, *p* < 0.001; Figure 6C). In addition to DWV levels, the NB KV GLMM revealed a positive association between KV and VDV1 (NB KV GLMM: VDV1 estimate = 0.85, *p* < 0.001; Figure 6D). This relationship was also seen in the NB VDV1 GLMM (NB VDV1 GLMM: KV estimate = 0.93, *p* < 0.001; Figure 6E). The NB KV GLMM also revealed a negative association between KV levels and the interaction between DWV and VDV1 levels (NB KV GLMM: DWV × VDV1 estimate = −0.83, *p* < 0.001; Figure 6D). Similarly, the NB VDV1 GLMM also revealed a negative association between VDV1 levels and the interaction between DWV and KV levels (NB VDV1 GLMM: KV × DWV estimate = −0.66, *p* < 0.001; Figure 6E).

## 4. Discussion

Here, we show that at least one *Varroa* mite is present in 90.4% of colonies sampled located in the southern Marmara region. It is important to point out that the sugar-roll method used here can recover up to 94% of *Varroa* mites from sampled nurse bees per colony [19]. Consequently, colonies with low levels of mites cannot be classified as *Varroa* mite-positive and mite counts from *Varroa* mite-positive colonies are 94% accurate. Therefore, it is possible that all colonies sampled are positive for *Varroa* mites and that mite counts are larger than what we found. Nevertheless, the prevalence of *Varroa* mites reported here is alarmingly higher than ones reported by two large-scale studies that examined *Varroa* mite prevalence across all regions of Turkey in different years [46,47], and a study that examined honey bee diseases in the southern Marmara region [48]. Two more recent large-scale studies identified a *Varroa* mite prevalence rate as low as 14.7% in Kırklareli [49] and as high as 100% [50] in Tekirdağ from the northern Marmara region. The increase in *Varroa* mite prevalence over the years has been also observed in Hatay and Adana from the Mediterranean region, where it was as low as 32% in 2003 [51] and as high as 100% in 2006–2007 [52,53]. This is similar to the increase seen in the Eastern Anatolia region, where it was as low as 25.6% between 2002–2004 in Elaziğ [54] and as high as 93–100% in Erzurum [55], Hakkari [56], and Kars [57]. The current *Varroa* mite prevalence, as well as its increase over the years, is consistent with that reported for other countries such as Estonia [58], Norway [59], Uruguay [60,61], and the US [62,63,64]. 

*Varroa* mites are active and passive vectors of more than a dozen bee-infecting viruses [65]. Therefore, we screened for those viruses using RNA sequencing (RNA-seq). Contrary to studies that reported a prevalence rate between 0–35.5% for ABPV, 20.2–32% for BQCV, 23–44.7% for DWV, 0–25% for CBPV, 0–6.5% for IAPV, 2.7–22.3% for SBV and 0% for KBV in different regions of Turkey [66,67,68,69]; we found alarmingly higher prevalence rate for ABPV (74.8%), BQCV (100%), DWV (100%), and absence of CBPV and IAPV. We detected SBV and KBV in 7% and 2.4% of the sampled colonies, respectively. Additionally, we report a prevalence rate of 56.5% for LSV, a virus that was very recently detected in *Varroa* mites sampled from İzmir and Muğla [70], 99.1% for KV, and 100% for VDV1. We suspect the distribution of viruses is likely due to the rise of *Varroa* mite infestations in Turkey because the viruses documented here, with nearly 100% prevalence, are the ones known to be vectored by *Varroa* mites and these could be competing and eliminating other possible viral infections once inside the host [64]. Although some sites have lower *Varroa* mite counts, they are still present as indicated by the high prevalence of *Varroa* mites for all sites. The lower number of mites appear to be sufficient for vectoring the viruses in a colony and because the viruses can be vertically transmitted [71], so it is likely that the viral population will grow once first introduced into a bee colony. In addition, except for ABPV, BQCV, DWV, KV, and VDV1, viral levels were very low, suggesting that those infections are covert (i.e., lack of symptoms with minimal or no effect on performance and lifespan). 

Stonebrood (SB) is a honey bee disease caused by *Aspergillus* spp., two of which, *A. flavus* and *A. fumigatus*, are the primary pathogens. SB is rare because natural *A. flavus* and *A. fumigatus* infections are unsuccessful, though they can occasionally multiply, in honey bee colonies [72]. Thus, SB is considered to be of minor importance to beekeepers. Here, we show that *Aspergillus* spp. That cause SB are present in 65.2% of colonies, from which nurse bees were examined, located in the southern Marmara region. Previous studies reported SB prevalence of 2%, 4.5%, 6.4%, and 14.3% in Erzurum from the Eastern Anatolia region [55], and Tekirdağ [73], Istanbul [74], and Kırklareli [49] from the northern Marmara region, respectively. This contrast between prevalence rates based on clinical symptoms of SB and the presence of *Aspergillus* spp. that cause SB is expected because *Aspergillus* spp. can be present in adult bees from non-SB-infected colonies. Alone, these *Aspergillus* spp. cannot establish a disease outbreak unless colonies are affected by multiple stressors (e.g., *Varroa* mite infestation, viral infections, etc.) that compromise individual and social immunity. However, it has been shown that *A. flavus* can overcome immune responses and establish an infection [75], so we assessed at least the potential for a SB outbreak within a colony. 

Chalkbrood (CB) is a honey bee brood disease caused by the fungi *Ascosphaera apis*, often regarded as an opportunistic pathogen. Both its outbreak and severity depends on a multitude of interacting stress-related factors, as in SB [76]. Adult bees, though responsible for spore transmission, are not susceptible to *A. apis*. The pathogen, however, can reside in adult bees [77,78] as nurse bees transmit the spore-contaminated food, in addition to being in close proximity to the infected brood, and removing the infected or dead larvae. Therefore, we screened for CB in nurse bees using PCR and found none of the sampled colonies to be infected. This is the first study that reports the absence of CB in the Marmara region, as many studies that examined CB prevalence located in the Marmara region between 2003–2013 reported a prevalence rate up to 36.3% [48,73,74,77]. However, our results are in concordance with that from a recent nation-wide study which reported a prevalence rate of 2.2% for CB [79]. The absence of CB could be due to the fact that we are sampling adult bees and it is more likely to be found infecting the brood. However, in general it is considered to be an opportunistic pathogen and generally not found at high levels within honey bee colonies [80].

American foulbrood (AFB) and European foulbrood (EFB) are two honey bee brood diseases caused by the bacteria *Paenibacillus larvae* and *Melissococcus plutonius*, respectively. Worker bees, though not susceptible, can be used for the monitoring of AFB and EFB [81,82]. We therefore screened for AFB and EFB in nurse bees using qPCR and found a prevalence rate of 31.3% and 6.1%, respectively, in sampled colonies from the southern Marmara region. A southern Marmara-region wide study examining AFB and EFB prevalence microbiologically in 2001 did not detect *P. larvae*, but detected *M. plutonius* in 5% of the sampled colonies [48]. An Istanbul-wide study examining honey samples with combs detected *P. larvae* and *M. plutonius* in 3.2% and 5.8% of the samples, respectively [74]. Both of these studies point out to the low prevalence of AFB and EFB compared to that reported by a large-scale study that found 29% and 19% prevalence for AFB and EFB, respectively, in Hatay and Adana between 2006–2007 [53]. The higher amounts of AFB is surprising given that this foulbrood is the one that is heavily managed because the bacteria is so persistent in the environment. Typically, these infections must be reported, the treatments are swift and dramatic to prevent its spread [83]. We did notice clinical signs of AFB when sampling for bees, the colonies had an odd odor with diseased brood, these colonies also appeared to have lower amounts of honey stored as well. 

*Nosema* is a globally prevalent adult honey bee disease caused by the two microsporidian parasite species *Nosema ceranae* and *Nosema apis*. In this study, we found none of the sampled colonies to be infected with *N. apis*, and *N. ceranae* in 64.3% of all sampled colonies located in the southern Marmara region. The *N. ceranae* prevalence rate reported here is higher than what was reported by previous studies in the Marmara region [46,50,73,84], but consistent with that reported by recent studies in the Aegean [85], Black Sea [86], and Eastern Anatolia region [87]. Our results are all also consistent with that reported from Belgium [88], Iran [89], Bulgaria [90], Canada [91,92], and the US [64]. 

Altogether, we show that the majority of colonies are infested with *Varroa* mites and infected with *N. ceranae*, *Aspergillus* spp., ABPV, BQCV, DWV, KV, and VDV1. Since *Aspergillus* spp. can only cause SB in colonies that are weakened by multiple stressors, it is possible for those colonies with *Varroa* mite infestation and multiple infections to develop symptoms of SB. Interestingly, we found that with the Marmara Island and Karacabey honey bee colonies there is an overall lower prevalence of pathogens, which suggests that with certain beekeeping practices and environmental conditions it is possible to reduce the honey bee disease burden in the Marmara region of Turkey. Marmara Island is isolated from other bee colonies, which may reduce horizontal transmission opportunities from other apiaries that result in increased disease loads (Fries and Camazine 2001). In addition, regular treatment for *Nosema ceranae* and *Varroa* mites have been conducted at this apiary location throughout the beekeeping season and subsequently these colonies have the lowest abundance of *Nosema ceranae* and *Varroa* mites in comparison to most of the other sampling locations. Because SB is positively correlated with *Varroa* mite levels this could explain why there are lower SB levels in the Marmara Island region as well. Given that the *Varroa* mite was so pervasive in comparison to the other bee pathogens and likely to be responsible for vectoring the pervasive of SB, ABPV, BQCV, DWV, KV, and VDV1, management practices geared towards keeping *Varroa* mite numbers in check for all beekeepers appears to be critical for maintaining honey bee colony health in Turkey. 

Overall, the pathogen loads for the bee colonies in Turkey appear to be increasing. For the non-viral infections, we found an interesting positive relationship between *Varroa* mites and SB. There could be a number of explanations for this relationship, but this supports the notion that *Varroa* mites can serve as a vector for this bee disease as well by transmitting SB spores horizontally from one individual to the next as this particular parasitic bee fungi has been found on the outside cuticle of *Varroa* mites [93]. As for viruses, ABPV was found to be negatively associated with *Nosema*, but positively associated with VDV1. Our findings support the notion that there might be a complex double-repressor relationship between ABPV/DWV/Varroa and LSV/Nosema. The viruses may be competing to use the same machinery to replicate and they may be in competition with LSV, which may facilitate a *Nosema ceranae* infection [64]. ABPV has been associated with *Varroa* mite infested colonies as it is vectored by this mite as well as VDV1, which is a likely explanation for why we observe the positive relationship between the two [94]. Therefore, ABPV could be key in facilitating the more virulent pathogen, VDV1, which is linked more directly to colony losses worldwide [95]. In contrast, BQCV did not interact with any other pathogens, but was very prevalent in all colonies sampled at low levels. Therefore, this parasite may act in parallel with other diseases in colonies that have multiple infections, which eventually results in colony collapse.

Our results suggest that DWV, KV, and VDV1 seem to form an interesting disease complex where if there is one other present, then levels are positively associated with each other, but if two other viruses are present within the complex, then there is a negative relationship. Therefore, even though these viruses are closely related to one another and can potentially recombine [96], they may be competing with one another, causing them to take independent evolutionary trajectories to increase their fitness levels. The positive relationship between the presence of one other virus within the complex highlights the importance of reducing DWV and KV infections even though these may not be as virulent as the VDV1 virus because their presence can increase the chances and disease load of honey bees being infected with the more virulent VDV1 [97,98]. When both viruses are present, then the amount of VDV1 is lower, but this probably is due to the general competition of resources within the host from three different viruses and from multiple infections the health of the bee may be severely affected [99]. It is interesting to note that although KV was not positively associated with *Varroa* mites, it was prevalent and played a role in the main interactions among the DWV and VDV1 bee viruses found in the honey bee colonies. 

In summary, as evidenced here, bee diseases are widespread and abundant throughout the Marmara region of Turkey. *Varroa* mites appear to be a central problem for beekeeping in Turkey because as shown here they are associated with prevalence and high loads of viral infections as well as the opportunistic parasitic fungi, SB. More research is needed to understand how the prevalent viruses are interacting with one another inside the bee host and how this translates to increased virulence and bee mortality such that a colony might collapse. Nonetheless, here we present documentation of one aspect that has played a major role in the worldwide bee health declines, which may lead to further investigation of how these prevalent diseases may lead to colony collapse in Turkey.

## Figures and Tables

**Figure 1 vetsci-09-00573-f001:**
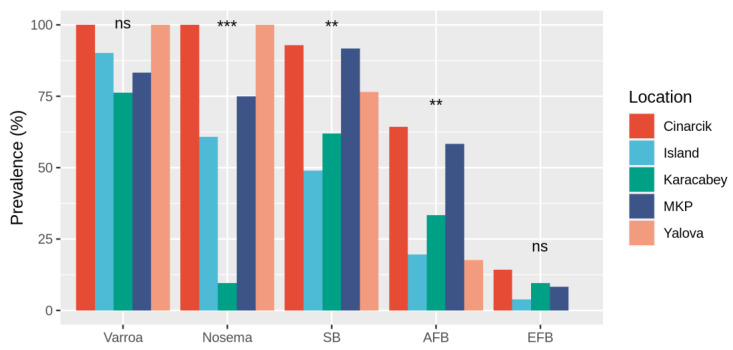
The prevalence of Varroa mites (Varroa), *Nosema ceranae* (Nosema), stonebrood (SB), American foulbrood (AFB), and European foulbrood (EFB) in terms of a percentage found at a colony level. Each color represents a different sampling apiary location within the Marmara region of Turkey. Statistical significance was determined using Fisher’s exact test. *** *p* ≤ 0.001, ** *p* ≤ 0.01, not significant (ns) *p* > 0.05.

**Figure 2 vetsci-09-00573-f002:**
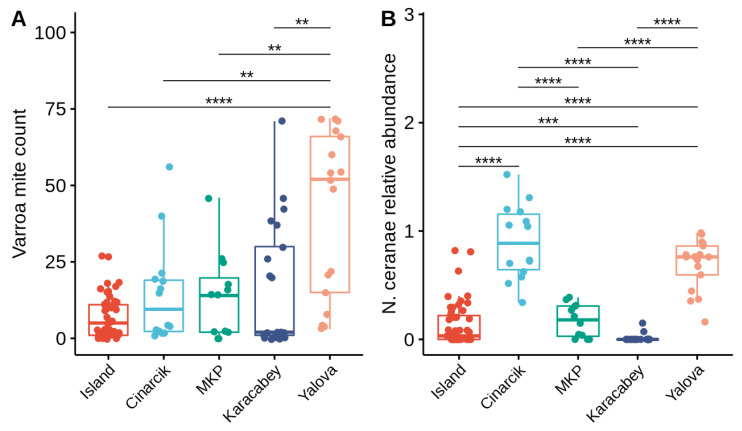
Median and inter-quartile range differences in *Varroa* mite count (**A**) and *N. ceranae* relative abundance (**B**) of colonies sampled from the five different apiary locations. Each color corresponds with a different sampling location. The sample size (n) is indicated below each box plot. Pairwise comparisons were done using Dun’s test and only significant differences were shown. **** *p* ≤ 0.0001, *** *p* ≤ 0.001, ** *p* ≤ 0.01.

**Figure 3 vetsci-09-00573-f003:**
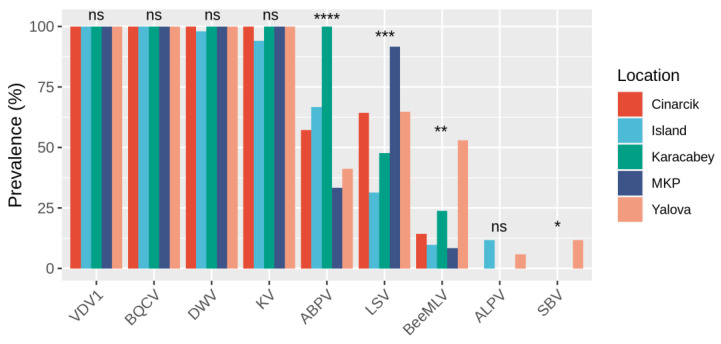
Prevalence of 9 out of the 14 honey bee viruses screened for that were detected in appreciable amounts. Each color represents one of the five apiary sampling locations. Statistical significance was determined using Fisher’s exact test. **** *p* ≤ 0.0001, *** *p* ≤ 0.001, ** *p* ≤ 0.01, * *p* ≤ 0.05, not significant (ns) *p* > 0.05.

**Figure 4 vetsci-09-00573-f004:**
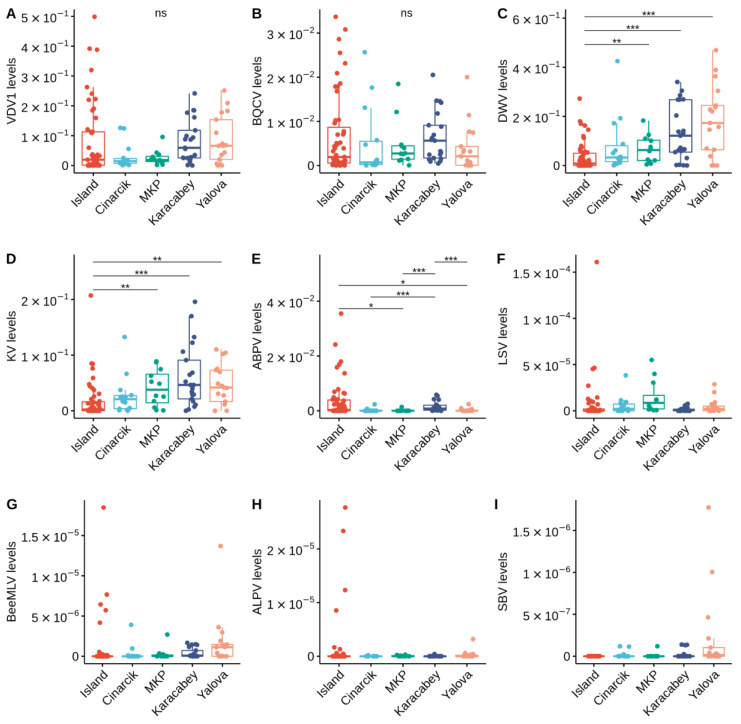
Relative abundance (viral reads/total reads) of VDV1 (**A**), BQCV (**B**), DWV (**C**), KV (**D**), ABPV (**E**), LSV (**F**), BeeMLV (**G**), ALPV (**H**), and SBV (**I**). Each color represents one of the five apiary sampling locations. Statistical significance was determined using the Kruskal–Wallis test followed by Dunn’s test for VDV1, BQCV, DWV, KV, and ABPV. Only significant differences were shown. *** *p* ≤ 0.001, ** *p* ≤ 0.01, * *p* ≤ 0.05, not significant (ns) *p* > 0.05.

**Figure 5 vetsci-09-00573-f005:**
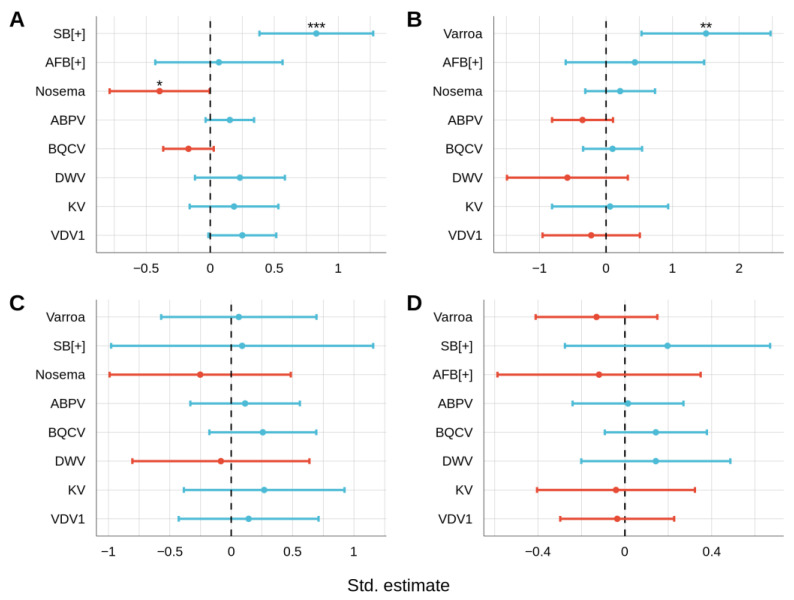
Standardized estimates from the GLMM analysis of Varroa mites (**A**), SB (SB) (**B**), American foulbrood (AFB) (**C**), and *N. ceranae* (**D**). Green lines indicate 95% confidence intervals that have an overall positive relationship while red lines indicate a negative relationship. * *p* < 0.05, ** *p* < 0.01 and *** *p* < 0.001.

**Figure 6 vetsci-09-00573-f006:**
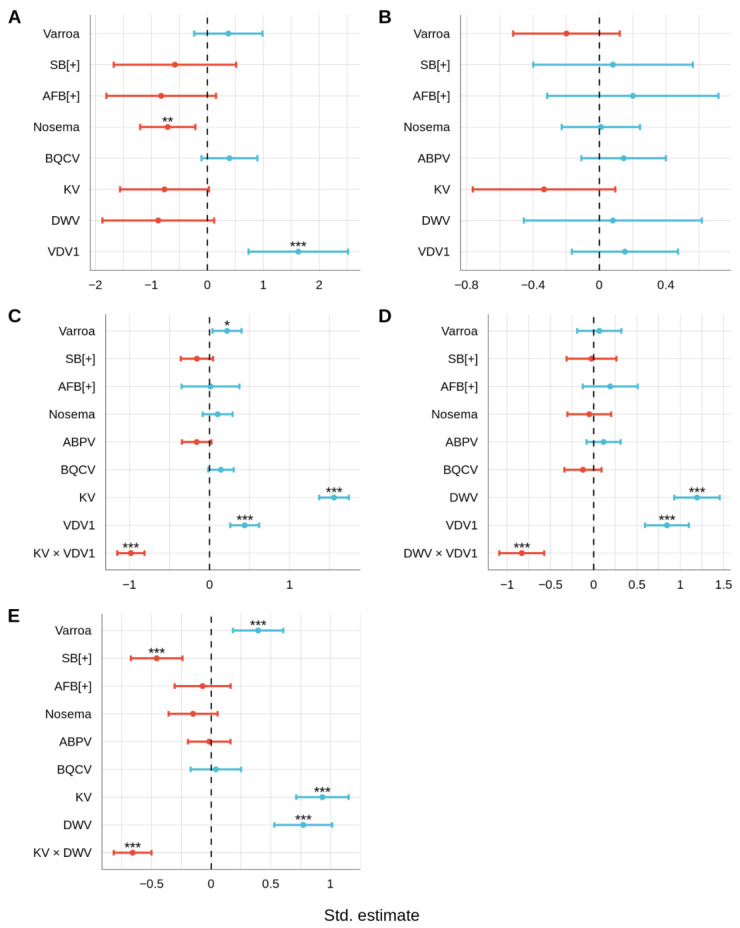
Standardized estimates from the GLMM analysis of Acute Bee Paralysis Virus (ABPV) (**A**), Black Queen Cell Virus (BQCV) (**B**), deformed wing virus (DWV) (**C**), Kakugo Virus (KV) (**D**), and Varroa Destructor Virus-1 (VDV1) (**E**). Green lines indicate 95% confidence intervals that have an overall positive relationship, while red lines indicate a negative relationship. * *p* < 0.05, ** *p* < 0.01 and *** *p* < 0.001.

**Table 1 vetsci-09-00573-t001:** The date, name, location, and number of colonies sampled of the five different apiaries within the Marmara region of Turkey during 2020 and 2021 for pathogen screening.

Date	Name of Location	GPS Coordinates	Sample Size
December 2020	Marmara Island (Island)	40.6227° N, 27.6175° E	19
July 2021	Marmara Island (Island)	40.6227° N, 27.6175° E	32
	Karacabey	40.2160° N, 28.3590° E	21
	Mustafakemalpasa (MKP)	40.0394° N, 28.4052° E	12
	Cinarcik	40.6452° N, 29.1192° E	14
	Yalova	40.6549° N, 29.2842° E	17

## Data Availability

The data and scripts used for the analysis for this study are publicly available at: https://github.com/hasimhko/tr-honeybee-disease-prev (accessed on 1 September 2022).

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
