# Peer review of "Honey Bee Pathogen Prevalence and Interactions within the Marmara Region of Turkey"

_vetsci, 2022, doi:10.3390/vetsci9100573_

Round 1

Reviewer 1 Report

In their research article, Mayack et al list honey bee diseases which they identified in the Marmara region, Turkey, as an attempt to explain why the Turkish beekeeping sector may be less developped than in other countries. They detected multiple pathogens using PCR and RNA-seq methods. However, they do not bring any kind of proof that the observed pathogens may indeed explain the low productivity of Turkish colonies: answering this type of question would require comparisons with other countries, and for instance include other socioeconomic aspects. The list of pathogens identified presents no novelty, given the general knowledge of their global distribution. The calculations of correlations between the occurences of the pathogens does not bring any kind of answer to the original problem. Overall, the scientific soundness of this article is very low and I would recommend rewriting it completely perhaps as a short note, just citing the identified diseases, but I am not even sure that this would bring anything to the scientific community.

Author Response

Dear reviewer,

We would like to thank you for the detailed comments and feedback on the manuscript entitled “Honey bee disease prevalence and interactions within the Marmara region of Turkey”. Your critical insights were very valuable to the development and improvement of the manuscript. We realize that the reviewer process is voluntary and appreciate the time you have taken to review this manuscript. 

We have reduced the results section, re-worked in the introduction, and have added substantial content to the discussion section in relation to the interpretation of our results. We have toned down the connection between diseases prevalence and low honey yield from Turkish bee colonies in a number of sections throughout the manuscript.

Below you will find a point-by-point response for the remaining issues that have been raised, our responses are indicated by the italicized text. All changes were recorded in the main manuscript file using track changes in Word. With all of these comments addressed we now feel that our manuscript is improved.

Sincerely,

Hasim Hakanoglu

In their research article, Mayack et al list honey bee diseases which they identified in the Marmara region, Turkey, as an attempt to explain why the Turkish beekeeping sector may be less developped than in other countries. They detected multiple pathogens using PCR and RNA-seq methods. However, they do not bring any kind of proof that the observed pathogens may indeed explain the low productivity of Turkish colonies: answering this type of question would require comparisons with other countries, and for instance include other socioeconomic aspects. The list of pathogens identified presents no novelty, given the general knowledge of their global distribution. The calculations of correlations between the occurences of the pathogens does not bring any kind of answer to the original problem. Overall, the scientific soundness of this article is very low and I would recommend rewriting it completely perhaps as a short note, just citing the identified diseases, but I am not even sure that this would bring anything to the scientific community.

We agree with the review that in this study we cannot explicitly address the connection between disease prevalence and low honey production in Turkey. We have therefore re-written segments in the summary, abstract and main text to make it clear that we are unable to test this hypothesis directly and instead we are providing indirect support for this idea. 

However, it is not true that the general knowledge of the pathogens' global distribution is known of some of the pathogens measured in this study. Very very few or no studies on the prevalence of ALPV, BSRV, BeeMLV, and KV were published. Some region-specific studies looking into the distribution of LSV and VDV-1 were published, but nothing on its global distribution (see Beaurepaire A, Piot N, Doublet V, et al. Diversity and Global Distribution of Viruses of the Western Honey Bee, Apis mellifera. Insects. 2020;11(4):239. Published 2020 Apr 10. doi:10.3390/insects11040239). Most of these studies are looking into prevalence only, nothing about viral levels. Also, we are using RNA-seq to reach a level of sensitivity and specificity beyond that of any qPCR assay.

Reviewer 2 Report

In general, manuscript is well written, lot of measurements is done, really nice work! The Introduction and Methods part are very well written. Results in my opinion should be shortened while the major concern comes with discussion part. You can find specific comments below.

Materials and Methods

In general, very nicely written. At the beginning please provide some general information: when the samples were collected (at least month and year), which locations were used and what type of hives are used. Through the text, use standardized name of location. Define this in methods part and use these names all the way.

Results

Result section is too long; specifically, it is not necessary to report all the results in the text if they are presented in the graph/table. Please point out only interesting situations in the text, without repeating everything that is shown in figures.

Discussion

You compare results nicely to the other studies; however, I am missing your opinion what is the reason that some of your results are different in your study? What could be the reason? For example, you found alarmingly higher prevalence rate for some viruses (L507-509) comparing to the previous studies. Why is that? Or, you write that “This is the first study that reports the absence of CB in the Marmara region”. What is your explanation on this? AFB is found in 31,3 % of colonies. Are there some clinical symptoms or this is something that is normally expected? The meaning of the discussion part is not only to compare results with other studies but to provide constructive discussion and explanation what are the reasons that lays behind your results. In this sense, the whole discussion part should be much more improved.

Some general questions that should be considered in discussion:

-       You find a presence of most of diseases in colonies. However, did you observe some clinical symptoms in the colonies?

-       It is interesting to see high prevalence of AFB on at least two apiaries, were there some reports from sampling apiaries on symptoms?

-       Why is prevalence of EFB much higher in two apiaries?

-       Why is SB much lower in Island location?

-       What do you thing, why there is no difference in virus prevalence of VDV1, DWV, BQCV and KV between location despite the fact that some apiaries were significantly less infected with varroa?

In the most part of the text you should change hive and beehive to colony and honey bee colony. Hive is a wooden box, colony is the living organism

Specific comments:

L39 – Apis mellifera in brackets after honey bee

L41 – food safety instead of food security

L50 – please try to provide reference for “Turkish Statistical Institute [TUIK]” or the source where this information could be found

L52-53 – please put in the reference list the FAO report. This goes for all FAO reports that you cite.

L58-59 – this statement is not very accurate because there are many factors that affect the productivity of honey. I wouldn’t say that diseases are the major problem in honey production per hive, as beekeepers in Canada, USA and other countries also face the same problems with diseases as beekeepers in Turkey.

L60 – Instead of “there are a number of factors” rewrite “there is a number of factors” or “there are many factors”.

L80 – honey bee colonies instead of hives

L88-92 – these are results and should not be here. Instead, please explain what is the aim of the study.

L93 – before sample collection subchapter or at the beginning of this subchapter, please explain the timing and location the samples collection: at which time samples were collected, from how many apiaries, how many colonies per region or per apiary. The date or month of sample collection are very important when reading results.

L99-101 – in which way the infestation of bees with mites was calculated? You assumed there are 300 bees in a sample, you weighed the samples?

L103 – from which part of the hive the bees were collected? Brood chamber or honey chamber?

L297-L321 – it is not necessary to report each result in the text as they are all shown on figure 1, it makes it hard to read. Instead, highlight which results are most interesting and show significant differences if any.

L498-503 – provide references for this. Also, there is no need to list all the viruses here, instead you can simple end the sentence after “viruses” in line 499 and put a few references where is described which viruses are vectored.

L507-509 – why is this in your study different than in previous ones? What is the reason do you know this or suspect on something?

L538-540 – “This is the first study that reports the absence of CB in the Marmara region”. What is the reason for this? Please discuss about it.

L546-547 – what does the prevalence of 31,3 % of AFB means for beekeeping? Were there some colonies with clinical symptoms? Are these colonies where AFB is determined sick or healthy? Is this affecting the honey production as you stated in the Introduction part?

L570-573 – what beekeeping practices are used in Island and how they differ from the other regions?

L570-571 – you mentioned here for the first time Karacabey honey bee hives. What kind of hives are these? If different types of hives were used in different apiaries this should be documented in M&M part.

L580 – please provide some explanations of this relationship if there are number of them.

L585-586 – explain hoe ABPV could be key in facilitating the VDV1? A relationship that you find does not have to mean that it is causal!

L591-596 – you should be really careful with this statement here. Your study concept is not allowing this kind of conclusion and I suggest to delete this!

L590 – why you didn’t include the BQCV in this complex and in the model?

L596-599 – please provide references for this.

L600-602 – in the reference that you placed here, this is shown for competition between DWV and SBV/BQCV, however not for VDV1-KV-DWV what you are talking about in lines 600-602. Please provide the correct reference for this.

L604-605 – provide reference for this that there is an evidence about this competition.

L606-609 – this is totally wrong conclusion! Yes, you shown that diseases are present in all places. However, you didn’t measure the honey production, bee population and brood amount in these 115 colonies so you can’t conclude that presence of diseases are the reason. For example, I can run a study in the XY country with much lower income than USA and find there that diseases are present in XY country. But also, the same diseases are found in USA. Do you think that is right to conclude that diseases are the cause of much lower income in XY?

Author Response

Dear reviewer,

We would like to thank you for the detailed comments and feedback on the manuscript entitled “Honey bee disease prevalence and interactions within the Marmara region of Turkey”. Your critical insights were very valuable to the development and improvement of the manuscript. We realize that the reviewer process is voluntary and appreciate the time you have taken to review this manuscript. 

We have reduced the results section, re-worked in the introduction, and have added substantial content to the discussion section in relation to the interpretation of our results. We have toned down the connection between diseases prevalence and low honey yield from Turkish bee colonies in a number of sections throughout the manuscript.

Below you will find a point-by-point response for the remaining issues that have been raised, our responses are indicated by the italicized text. All changes were recorded in the main manuscript file using track changes in Word. With all of these comments addressed we now feel that our manuscript is improved.

Sincerely,

Hasim Hakanoglu

In general, manuscript is well written, lot of measurements is done, really nice work! The Introduction and Methods part are very well written. Results in my opinion should be shortened while the major concern comes with discussion part. You can find specific comments below.

We thank the reviewer for the positive feedback!

Materials and Methods

In general, very nicely written. At the beginning please provide some general information: when the samples were collected (at least month and year), which locations were used and what type of hives are used. Through the text, use standardized name of location. Define this in methods part and use these names all the way.

We have now added in these details at the beginning of the methods section and have made our naming system consistent throughout the manuscript.

Results

Result section is too long; specifically, it is not necessary to report all the results in the text if they are presented in the graph/table. Please point out only interesting situations in the text, without repeating everything that is shown in figures.

We agree with the reviewer and have shortened the results section. This information is now included on the graphs and the significant differences are indicated with a *.

Discussion

You compare results nicely to the other studies; however, I am missing your opinion what is the reason that some of your results are different in your study? What could be the reason? For example, you found alarmingly higher prevalence rate for some viruses (L507-509) comparing to the previous studies. Why is that? Or, you write that “This is the first study that reports the absence of CB in the Marmara region”. What is your explanation on this? AFB is found in 31,3 % of colonies. Are there some clinical symptoms or this is something that is normally expected? The meaning of the discussion part is not only to compare results with other studies but to provide constructive discussion and explanation what are the reasons that lays behind your results. In this sense, the whole discussion part should be much more improved.

We agree with the reviewer that the discussion could benefit from more interpretation and speculation based on our findings. We have now added in this content to the discussion section.

Some general questions that should be considered in discussion:

-       You find a presence of most of diseases in colonies. However, did you observe some clinical symptoms in the colonies?

-       It is interesting to see high prevalence of AFB on at least two apiaries, were there some reports from sampling apiaries on symptoms?

-       Why is prevalence of EFB much higher in two apiaries?

This was actually a mistake in the figure and it has now been corrected.

-       Why is SB much lower in Island location?

-       What do you thing, why there is no difference in virus prevalence of VDV1, DWV, BQCV and KV between location despite the fact that some apiaries were significantly less infected with varroa?

We thank the reviewer for raising these interesting points. We have considered these in our discussion section and added content accordingly.

In the most part of the text you should change hive and beehive to colony and honey bee colony. Hive is a wooden box, colony is the living organism

This correction has been made throughout the manuscript.

Specific comments:

L39 – Apis mellifera in brackets after honey bee

Corrected.

L41 – food safety instead of food security

Corrected.

L50 – please try to provide reference for “Turkish Statistical Institute [TUIK]” or the source where this information could be found

This has now been corrected.

L52-53 – please put in the reference list the FAO report. This goes for all FAO reports that you cite.

This has now been corrected.

L58-59 – this statement is not very accurate because there are many factors that affect the productivity of honey. I wouldn’t say that diseases are the major problem in honey production per hive, as beekeepers in Canada, USA and other countries also face the same problems with diseases as beekeepers in Turkey.

We have toned down the statement suggesting that there is an explicit connection between the two, this is a possible cause, but it is only suspected to be one of many potential causes.

L60 – Instead of “there are a number of factors” rewrite “there is a number of factors” or “there are many factors”.

Corrected.

L80 – honey bee colonies instead of hives

Corrected.

L88-92 – these are results and should not be here. Instead, please explain what is the aim of the study.

The results have been removed and the aim of the study has been put in its place.

L93 – before sample collection subchapter or at the beginning of this subchapter, please explain the timing and location the samples collection: at which time samples were collected, from how many apiaries, how many colonies per region or per apiary. The date or month of sample collection are very important when reading results.

These additional details have now been added into the methods section.

L99-101 – in which way the infestation of bees with mites was calculated? You assumed there are 300 bees in a sample, you weighed the samples?

We assumed that there were 300 bees per 1 cup of bees. This detail has been added into the manuscript.

L103 – from which part of the hive the bees were collected? Brood chamber or honey chamber?

The location of collection has been added to the manuscript.

L297-L321 – it is not necessary to report each result in the text as they are all shown on figure 1, it makes it hard to read. Instead, highlight which results are most interesting and show significant differences if any.

We have now reduced this part of the results section. 

L498-503 – provide references for this. Also, there is no need to list all the viruses here, instead you can simple end the sentence after “viruses” in line 499 and put a few references where is described which viruses are vectored.

This sentence has been re-worked.

L507-509 – why is this in your study different than in previous ones? What is the reason do you know this or suspect on something?

We suspect that this is due to the rise in Varroa mite infestations in recent years because the viruses with highest prevalence are the ones vectored by Varroa mites. We have added this into the discussion section.

L538-540 – “This is the first study that reports the absence of CB in the Marmara region”. What is the reason for this? Please discuss about it.

Some speculation on the reason for this has been added into the manuscript.

L546-547 – what does the prevalence of 31,3 % of AFB means for beekeeping? Were there some colonies with clinical symptoms? Are these colonies where AFB is determined sick or healthy? Is this affecting the honey production as you stated in the Introduction part?

Some of the bee colonies has clinical symptoms of AFB as they were producing an odd odor and had infected brood with signs of AFB. These colonies appeared to be sick and had lower honey yields in comparison to the non infected hive.

L570-573 – what beekeeping practices are used in Island and how they differ from the other regions?

Some speculation on why this area may have lower disease prevalence in comparison to the others has been added into the manuscript.

L570-571 – you mentioned here for the first time Karacabey honey bee hives. What kind of hives are these? If different types of hives were used in different apiaries this should be documented in M&M part.

These hive types were the same as the others.

L580 – please provide some explanations of this relationship if there are number of them.

Additional explanations and references have now been provided.

L585-586 – explain hoe ABPV could be key in facilitating the VDV1? A relationship that you find does not have to mean that it is causal!

L591-596 – you should be really careful with this statement here. Your study concept is not allowing this kind of conclusion and I suggest to delete this!

We have toned down this statement so that it is more in line with our study design.

L590 – why you didn’t include the BQCV in this complex and in the model?

It was included in the analysis but no significant interactions with other viruses came out of the analysis so we assumed it does not take part in the potential disease complex of interacting viruses.

L596-599 – please provide references for this.

References have been added in to support this claim.

L600-602 – in the reference that you placed here, this is shown for competition between DWV and SBV/BQCV, however not for VDV1-KV-DWV what you are talking about in lines 600-602. Please provide the correct reference for this.

The sentence has been re-worked so that it is more clear to the reader of what we were trying to communicate.

L604-605 – provide reference for this that there is an evidence about this competition.

We could not find a reference for this speculation and instead it is based on the negative correlation we observed from our analysis. We have therefore removed this statement from the discussion section.

L606-609 – this is totally wrong conclusion! Yes, you shown that diseases are present in all places. However, you didn’t measure the honey production, bee population and brood amount in these 115 colonies so you can’t conclude that presence of diseases are the reason. For example, I can run a study in the XY country with much lower income than USA and find there that diseases are present in XY country. But also, the same diseases are found in USA. Do you think that is right to conclude that diseases are the cause of much lower income in XY?

We agree with the reviewer. This sentence has been re-worked so that the connection is not so explicit, and the statement has been toned down.

Reviewer 3 Report

This paper provides an extensive diagnostic effort to assess the health status of honeybee colonies in the Marmara region of Turkey. The amount of results provides a comprehensive picture of the current situation, point out that there is a urgent need for actions to avoid even more severe losses in the future.

Some line by line comments to improve the manuscript are listed below:

L10: replace lower with low or add a comparative term

L16 and L26 replace diseases with pathogens

L27-28: this sentence should be rephrased, it now sounds as 100% of bees were diseased, whereas you mean that disease was present in 100% of hives, but we don’t know the prevalence among bees.

L32-33 this sentence is unclear and should be rephrased

Introduction: How the references from TUIK and FAO organizations can be fully retrieved? It seems that they are not included in the reference list.

L53, 56, 58: the reference to Fig.1 looks unproperly mentioned here, there is no panel A,B or C in Fig.1!

L76-79: In this paragraph the authors should add references to the current state of the art regarding the parasites and pathogens of honeybee whose presence has been previously in Turkey.

L101-102. A map of sampling sites should be provided, or alternatively a table with GPS coordinates, and the number of hives per sites should be provided.

L103-104 How were the bees selected and collected?

In section 2.3 I recommend to avoid the confusion between pathogen and disease, the reported molecular tools test for pathogen infection, not for disease presence (i.e., symptom appearance).

2.3.4 The procedure is not clear. Why DNA extraction was performed twice?

2.4 Statistical analysis: Please specify that for prevalence calculation separate analyses were made for different sites, to obtain a picture of different prevalence in different areas.

L320 replace very low with “the lowest”

Fig.1. It looks like there is an error in the graph of EFB, since the Karacabey and MKP columns are very tall. Moreover, providing letters for different values will be helpful for the reader.

In the discussion the authors could improve the discussion on possible causes for viral fluctuations with respect to reference data.

Author Response

Dear reviewer,

We would like to thank you for the detailed comments and feedback on the manuscript entitled “Honey bee disease prevalence and interactions within the Marmara region of Turkey”. Your critical insights were very valuable to the development and improvement of the manuscript. We realize that the reviewer process is voluntary and appreciate the time you have taken to review this manuscript. 

We have reduced the results section, re-worked in the introduction, and have added substantial content to the discussion section in relation to the interpretation of our results. We have toned down the connection between diseases prevalence and low honey yield from Turkish bee colonies in a number of sections throughout the manuscript.

Below you will find a point-by-point response for the remaining issues that have been raised, our responses are indicated by the italicized text. All changes were recorded in the main manuscript file using track changes in Word. With all of these comments addressed we now feel that our manuscript is improved.

Sincerely,

Hasim Hakanoglu

This paper provides an extensive diagnostic effort to assess the health status of honeybee colonies in the Marmara region of Turkey. The amount of results provides a comprehensive picture of the current situation, point out that there is a urgent need for actions to avoid even more severe losses in the future.

We thank the reviewer for the positive feedback!

Some line by line comments to improve the manuscript are listed below:

L10: replace lower with low or add a comparative term

Correction made.

L16 and L26 replace diseases with pathogens

Correction made.

L27-28: this sentence should be rephrased, it now sounds as 100% of bees were diseased, whereas you mean that disease was present in 100% of hives, but we don’t know the prevalence among bees.

We agree with the reviewer, this sentence has been re-worked to make it more clear to the reader.

L32-33 this sentence is unclear and should be rephrased

This sentence has been re-worked to make it more clear to the reader.

Introduction: How the references from TUIK and FAO organizations can be fully retrieved? It seems that they are not included in the reference list.

These have now been added into the reference list.

L53, 56, 58: the reference to Fig.1 looks unproperly mentioned here, there is no panel A,B or C in Fig.1!

These were accidentally included and now have been removed, thank you for pointing this out.

L76-79: In this paragraph the authors should add references to the current state of the art regarding the parasites and pathogens of honeybee whose presence has been previously in Turkey.

Content plus a reference was added 

L101-102. A map of sampling sites should be provided, or alternatively a table with GPS coordinates, and the number of hives per sites should be provided.

A table, as suggested, has been added to the methods section.

L103-104 How were the bees selected and collected?

Bees from the brood area were collected, the same bees that were sampled for Varroa mite counts were used for pathogen screening. This information has been added to the methods section.

In section 2.3 I recommend to avoid the confusion between pathogen and disease, the reported molecular tools test for pathogen infection, not for disease presence (i.e., symptom appearance).

This has now been corrected.

2.3.4 The procedure is not clear. Why DNA extraction was performed twice?

DNA extraction was performed twice because different extraction procedures can lead to different proficiency levels in terms of DNA yield from the pathogen of interest. For example, in order to obtain AFB DNA a high temperature is needed to burst open the cell because it has a tough spore like membrane in comparison to EFB.

2.4 Statistical analysis: Please specify that for prevalence calculation separate analyses were made for different sites, to obtain a picture of different prevalence in different areas.

This information was added to this section.

L320 replace very low with “the lowest”

Correction made.

Fig.1. It looks like there is an error in the graph of EFB, since the Karacabey and MKP columns are very tall. Moreover, providing letters for different values will be helpful for the reader.

The reviewer is correct, this has now been corrected. We have also added in * to the graphs for significant differences resulting from the multiple comparisons.

In the discussion the authors could improve the discussion on possible causes for viral fluctuations with respect to reference data.

We agree with the reviewer and content has been added to the discussion section to address this point.

Round 2

Reviewer 2 Report

The authors have answered all the raised questions from the first review.

Author Response

We thank the reviewer for the feedback.